# Composite Gel Fabricated with Konjac Glucomannan and Carrageenan Could Be Used as a Cube Fat Substitute to Partially Replace Pork Fat in Harbin Dry Sausages

**DOI:** 10.3390/foods10071460

**Published:** 2021-06-24

**Authors:** Jiaxin Chen, Jinhai Zhao, Xin Li, Qian Liu, Baohua Kong

**Affiliations:** 1College of Food Science, Northeast Agricultural University, Harbin 150030, China; chenjiaxin993523@hotmail.com; 2Institute for Advanced Technology, Heilongjiang Academy of Sciences, Harbin 150001, China; zjh345729635@hotmail.com; 3Sharable Platform of Large-Scale Instruments & Equipments, Northeast Agricultural University, Harbin 150030, China; lixin810910@hotmail.com; 4Heilongjiang Green Food Science & Research Institute, Harbin 150028, China

**Keywords:** Harbin dry sausage, back-fat partial substitution, cube fat substitute, physicochemical characteristic, sensory evaluation

## Abstract

The effect of the partial replacement of pork back-fat with a cube fat substitute (CFS) fabricated from konjac glucomannan and carrageenan on the physicochemical and sensory characteristics of Harbin dry sausages during 7 days of fermentation was investigated. There were the following five treatments: control (100% back-fat), FS1 (80% back-fat, 20% CFS), FS2 (60% back-fat, 40% CFS), FS3 (40% back-fat, 60% CFS) and FS4 (20% back-fat, 80% CFS). The results showed no significant differences (*p* > 0.05) in the physicochemical and sensory characteristics among the control, FS1 and FS2 treatments. However, higher replacement levels (60% and 80%) rendered higher degrees of change in the characteristics of the sausages, lowering the moisture content and *a*_w_ and increasing the pH, hardness, chewiness and atypical appearance at the end of fermentation. Moreover, electronic nose analysis and hierarchical cluster analysis demonstrated that the FS3 and FS4 treatments destroyed the characteristic quality of the sausage. Overall, our results indicated that, to ensure the traditional characteristics of Harbin dry sausages, the upper limit of the fat replacement level with CFS should be set at 40%.

## 1. Introduction

Harbin dry sausage is characterised by its unique texture and flavour and is the most popular traditional fermented meat product in Northeast China [1]. The sausage is typically prepared by cube pork back-fat meat and lean meat mixed with spices, stuffed in casings and fermented for 4–7 days in a chamber at constant temperature and humidity. The average pork back-fat content in the dry sausages can be as high as 30% by the end of the fermentation process. It is generally known that animal fats confer favourable quality and acceptability properties (such as aroma, texture and mouthfeel) to fermented dry sausages [2]. However, research has proven that excessive fat intake is the main reason for an increased risk of cardiovascular and cerebrovascular diseases, obesity and some types of cancer [3,4,5]. Dietary guidelines prepared by the World Health Organization [6] state that consumers need to be aware of the dangers of excessive fat intake and daily fat intake should not exceed 30% of the total calories. The Chinese government has also implemented a series of strategies to protect consumers’ health. For instance, the “Chinese Dietary Guidelines” suggest that daily fat intake should be between 20 and 25% of the total calories [7]. Thus, it is necessary to reduce the pork back-fat content of Harbin dry sausages to ensure the healthy diet of consumers.

Reducing the fat content in fermented dry sausages is an enormous challenge faced by meat product industries worldwide. The most direct method and the method proposed by the meat products industry and governments is to reduce the fat addition level. Nevertheless, directly reducing the added fat is not feasible in fermented dry sausage products, because, quite apart from its economic value and nutritional contribution, the quality and acceptability properties (e.g., flavour, texture, mouthfeel) of the dry sausages will also be damaged [2]. Muguerza et al. [8] showed that reducing the fat level could increase the weight loss, hardness and firmness of fermented dry sausages and cause them to appear darker and redder. Liaros et al. [9] indicated that directly reducing the fat level from 30% to 10% could induce case hardening, deteriorate the external appearance and increase the lipid oxidation of dry sausages. Olivares et al. [10] also stated that excessive fat reduction in fermented dry sausages could affect the texture by increasing the chewiness and hardness and deteriorating the sensory acceptability.

Several studies have attempted to overcome these issues by using plant-based ingredients formed from vegetable oils and typical neutral polysaccharide gels (such as konjac gel, carrageenan, carboxymethylcellulose, Jerusalem artichoke powder, locust bean gum and xanthan gum) to partially reduce the fat content in these meat products [2,11,12]. The plant-based ingredients are rich in dietary fibre and have no caloric value, which is more conducive to the health of consumers [13]. Using the plant-based ingredients instead of animal fat may be beneficial for resource-intensive meat product manufacturers to reduce the environmental problems. More importantly, the plant-based ingredients own a high water-retention capacity that can facilitate the regular moisture release occurring during the fermentation process, ensuring the sensory properties of fermented dry sausages [14]. Alejandre et al. [15] concluded that replacing fat with a carrageenan gel resulted in fermented dry sausages with similar taste and juiciness to the traditional ones, without inducing oxidation problems. Jiménez-Colmenero et al. [16] indicated that the appearance, flavour and juiciness were acceptable in low-fat fermented dry sausages prepared with appropriate levels of a konjac gel matrix. Additionally, the gelatinous substitutes can be shaped into cubes and provide the appearance of the visible fat cubes required for use as a raw material to replace animal fats [17].

Konjac glucomannan is a neutral polysaccharide extracted from *Amorphophallus konjac*, a native plant of East Asia. Several studies have proved that it forms gels that, combined with carrageenan, can be successfully used as ‘fat analogues’ in the formulation of low-fat meat products [2]. Moreover, in our previous study, the best physicochemical and sensory properties were found in the gel cube fat substitutes (CFS) prepared with konjac glucomannan and κ-carrageenan, and the CFS conferred sensory properties of juiciness and texture similar to that of fat [18]. Based on these results, the objective of this current study is to explore the effect of CFS as a fat replacer (at 20, 40, 60 and 80% back-fat replacement) on the physicochemical and sensory characteristics of low-fat Harbin dry sausages. The moisture content, water activity (*a*_w_), pH, thiobarbituric acid-reactive substances (TBARS), colour, texture profile analysis (TPA), electronic nose (E-nose) and sensory evaluation were analysed. Additionally, hierarchical cluster analysis (HCA) was performed among all the physicochemical characteristics of the dry sausages to identify similarities and differences among the sausages with or without CFS.

## 2. Materials and Methods

### 2.1. Materials

Fresh pork lean-meat and back-fat were obtained from the Carrefour supermarket (Harbin, China) and transported on ice to the meat science laboratory, although some research involving human or animal participants usually requires ethical approval, this protection is not extended to farmed animals. The visible fat and connective tissue on the lean pork were removed. The ingredients used to prepare CFS included corn germ oil (COFCO Co., Ltd., Harbin, Heilongjiang, China), konjac flour (glucomannan 83%, 120 mesh; Johnson Konjac Technology Co., Ltd., Wuhan, Hubei, China), κ-carrageenan flour (Jingxie Marine Technology Co., Ltd., Jinan, Shandong, China), barley β-glucan (Yuwei Biotechnology Co., Ltd., Beijing, China) and food-grade Na_2_CO_3_ (Zhenlemen Food Co., Ltd., Xuzhou, Jiangsu, China). Other additives, such as NaCl, sodium nitrite and flavouring, were also purchased from the Carrefour supermarket.

### 2.2. CFS Preparation

The CFS was prepared according to the method of Ruiz-Capillas et al. [2] with slight modifications. Briefly, 4.5 g of konjac flour, 4.5 g of κ-carrageenan flour and 5.0 g of barley β-glucan were mixed in 76.0 g of water, then stirred at a speed of 1500 rpm for 6 min using a stirrer (OS-Pro, KeXing Co., Ltd., Shanghai, China) with a 5 min intermittence time. After that, 10.0 g of corn germ oil was added to the mixture and stirred at 3000 rpm for 3 min. Then, 10.0 mL of 3.0% (*w*/*v*) Na_2_CO_3_ solution was added to the mixture and stirred again under the same conditions. Finally, the mixture was placed in a suitable container to form CFS. The container was covered and manually compacted to remove air and then heated at 90 °C in a water bath for 60 min. The prepared CFS was stored at 4 °C until use.

### 2.3. Harbin Dry Sausage Preparation

The Harbin dry sausages were prepared according to the process described by Chen et al. [19] with slight modifications. Five different formulations of the dry sausages were prepared, as shown in Table 1, including a control group without CFS and the following four low-fat samples: FS1 (80% back-fat, 20% CFS), FS2 (60% back-fat, 40% CFS), FS3 (40% back-fat, 60% CFS) and FS4 (20% back-fat, 80% CFS). The sausage samples of each treatment were pre-dried at 40 ± 2 °C for 24 h and then transferred to an incubator for fermentation (25 ± 2 °C, 75–80% relative humidity). The sausage samples of each treatment were analysed at the designated fermentation times (day 0, 1, 4 and 7).

### 2.4. Moisture Content and a_w_

Moisture content was determined using the method of the AOAC [20]. Measurements of *a*_w_ were obtained using a water activity meter (Decagon Devices, Seattle, WA, USA), as described by Chen et al. [21].

### 2.5. pH and TBARS Values

The pH was determined according to the method of Sun et al. [22] with slight modifications. Briefly, the casing of the sausage sample was stripped, and approximately 10.0 g of sausage was stirred in 90.0 mL of distilled water at a speed of 500 rpm for 5 min using a stirrer (OS-Pro, KeXing Co., Ltd., Shanghai, China). The pH of the slurry was measured using a standard pH meter (Mettler Toledo Instruments Co., Ltd., Shanghai, China) at room temperature (25 °C).

Lipid oxidation was evaluated by measuring the TBARS using the method of Wen et al. [23]. Briefly, 2.0 g of minced sausage samples were weighed and mixed with 3.0 mL of thiobarbituric acid, followed by an addition of 17.0 mL 2.5% trichloroacetic acid. Then, the mixture was heated in boiling water for 30 min and cooled at room temperature. After that, 4.0 mL of suspension was mixed with the same volume of chloroform, and then centrifuged at 1800× *g* for 10 min. The supernatant was determined at 532 nm. Results were expressed as milligrams of malonaldehyde (MDA) per kilogram of sausage, and calculated using the following equation: TBARS (mg/kg) = *A*_532_/*ω* × 9.48where, among the equation, *A*_532_ is the absorbance (532 nm) of the assay solution, *ω* is the sample weight (g) and ‘9.48’ is a constant derived from the dilution factor and the molar extinction coefficient (152,000 M^−1^ cm^−1^) of the red thiobarbituric acid reaction product.

### 2.6. Colour Measurement

Instrumental colour was evaluated using the method of Chen et al. [5]. The samples of each treatment were maintained at 25 °C for 30 min and then sliced into approximately 2.0-centimeter-thick pieces. Colour values for the CIE colour coordinates *L** (lightness), *a** (redness) and *b** (yellowness) were measured using a ZE-6000 colourimeter (Nippon Denshoku, Kogyo Co., Tokyo, Japan).

### 2.7. Texture Profile Analysis

The TPA of the sausage samples was measured at 25 °C based on the method of Yin et al. [24] using a TA.XT2 plus Texture Analyser (Stable Micro Systems Ltd., Godalming, Surrey, UK) with a P-50 cylindrical probe. Four indexes were determined, including hardness (N), chewiness (N), springiness and resilience.

### 2.8. E-Nose Analysis

E-nose analysis was performed using the method of Yin et al. [25] using a PEN3 E-nose (Airsense Analytics GmbH, Schwerin, Germany). The information of 10 sensors for the E-nose is shown in Table 2.

### 2.9. Sensory Analysis

This study was registered and approved by the Ethics in Research Committee of Northeast Agricultural University (Harbin, China). The procedure for sensory evaluation was adopted from Kong et al. [26] with some modifications. Thirty sensory analysis panellists (15 females and 15 males) were selected in the meat science laboratory of Northeast Agricultural University due to their experience in the sensory evaluation of meat products. All of the panellists signed a consent form agreeing to participate as volunteers in the sensory analysis. Before the sensory evaluation, the experts from the meat science laboratory of Northeast Agricultural University conducted three preliminary, sample familiarisation training sessions for the 30 panellists, and a “warm-up” sample (cooked dry sausage slices) was submitted to every panellist to evaluate each sensory trait. Afterwards, the steamed sausage samples of each treatment were sliced into approximately 2-millimeter-thick pieces to begin the evaluation. The procedures of the sensory analysis were implemented according to a seven-point scale method [27] and carried out in a sensory laboratory designed in accordance with International Standard Organisation (ISO) [28]. Samples were evaluated for colour (1 = dark and dull, 7 = red and shiny), texture (1 = loose, 7 = compact), juiciness (1 = dry, 7 = juicy), firmness (1 = very hard, 7 = very soft), flavour (1 = extremely undesirable, 7 = extremely desirable) and acceptability (1 = low, 7 = high). Additionally, images of the different treatments, including uncooked sausages, cooked sausages and slices of cooked sausages containing true fat and CFS, were captured using a digital camera under the same conditions.

### 2.10. Statistical Analysis

All experiments were implemented in triplicate (triplicate observations) for each batch of sausages. Data were analysed using the General Linear Models procedure of Statistix 8.1 (Analytical Software, Saint Paul, MN, USA) and displayed as the mean ± standard error (SE). One-way analysis of variance (ANOVA) was used to assess the significance of the main effects (*p* < 0.05) between means using Tukey’s multiple comparison test. Figures were drawn using the Sigma Plot 13 software (Systat Software GmbH, San Jose, CA, USA). Three independent batches of sausages (replicates) were prepared. Principal component analysis (PCA) was performed among the treatments and sensors of E-nose analysis using SPSS Statistics version 22.0 (Analytical Software, New York, NY, USA). In addition, hierarchical cluster analysis (HCA) was performed among all physicochemical characteristics (moisture content, *a*_w_, pH, TBARS values, colour and TPA) of the dry sausages by using R software (version 3.6.3; Tsinghua University, Beijing, China) to identify similarities and differences among the sausages with or without CFS.

## 3. Results and Discussion

### 3.1. Moisture Content and a_w_

The moisture content and *a*_w_ gradually decreased in all the treatments during the 7-day fermentation process, as shown in Table 3. The initial moisture content was higher in all the treatment samples (FS1–FS4) than in the control (*p* < 0.05), and FS4 (with 80% CFS) recorded the highest moisture content (*p* < 0.05), perhaps due to the higher moisture content of the CFS than the back-fat. However, at the end of the pre-drying stage (24 h), all the treatment samples displayed a significantly lower moisture content compared to the control (*p* < 0.05). This moisture loss during pre-drying could be related to the high temperature (40 °C) accelerating the moisture evaporation [10].

After pre-drying, all the samples were fermented, during which further decreases in the moisture content occurred. At the end of fermentation (7 days), the moisture content had decreased from 62.52, 63.32, 64.35, 64.88 and 65.53% to 28.57, 27.31, 27.55, 24.36 and 22.06% in the control, FS1, FS2, FS3 and FS4 samples, respectively (*p* < 0.05). At this point, there was no significant difference among the control, FS1 and FS2 treatments (*p* > 0.05), but the FS3 and FS4 treatments had a significantly lower moisture content compared to the control (*p* < 0.05). The water holding capacity of the CFS is mainly related to the gel structure, which might have steadily worsened throughout the fermentation process, inducing a lower moisture content in the treatment samples with higher fat replacement levels (60% and 80%) [29]. The same results were also reported in low-fat dry fermented sausages amended with gelatinous substances [2,17]. Moreover, the water loss could not induce degradation of CFS in the fermentation process of dry fermented sausages, which means that it could potentially be used in non-fermented sausages.

The initial (day 0) *a*_w_ values were higher in the back-fat-reduced samples than in the control group (*p* < 0.05). All the samples showed a gradual decrease in *a*_w_ during the fermentation stage, reaching 0.84, 0.82, 0.81, 0.79 and 0.78 in the control, FS1, FS2, FS3 and FS4, respectively (*p* < 0.05), by the end of the fermentation process (7 days). The *a*_w_ of all the FS treatments was significantly lower than that of the control group at day 7, especially of the FS3 and FS4 treatments (*p* < 0.05). Although there was no linear correlation between moisture content and *a*_w_, both indexes showed a similar change trend [30]. The *a*_w_ is a colligative property that reflects the water state, mainly the amount of free (unbound) water. The lower the *a*_w_ is, the higher the degree of binding is. On the one hand, the texture of the dry sausages became more compact during the fermentation process, which may cause a higher degree of moisture-binding. On the other hand, the lower *a*_w_ in the FS3 and FS4 treatments could be related to higher moisture evaporation [9]. Similar changes in the *a*_w_ of dry fermented sausage have been reported by García et al. [31] and Mendoza et al. [32].

### 3.2. pH and TBARS Values

A rapid pH decline at the beginning of the fermentation stage is a typical characteristic of dry fermentation meat products that confers the unique flavour and enhances the quality of the product [33]. The pH changes of the dry sausages during the 7-day fermentation are shown in Figure 1. All the treatment samples exhibited a progressive decline in the pH throughout the fermentation stage. At day 0, the pH was higher in the reduced-back-fat samples than in the control group (*p* < 0.05), and it seemed that the higher the proportion of CFS in the dry sausages is, the higher the pH of the sample is, probably due to the higher pH of the CFS compared to that of the back-fat. As fermentation progressed, the pH of all the treatments rapidly dropped during the first few days and then decreased smoothly until the end of the fermentation process (day 7). It was notable that the highest pH was found in the FS4 treatment (*p* < 0.05) and was comparable to that of the FS3 treatment (*p* > 0.05). An excessively high pH is not a typical technical characteristic of Harbin dry sausages and could negatively affect consumer acceptability.

The TBARS are a reliable indicator of lipid oxidation in meat products that reflects the level of secondary oxidation products, such as malondialdehyde, aldehydes and ketones [34]. The TBARS data for all the treatments are presented in Figure 2. All the TBARS values were comparable at day 0 (*p* < 0.05) and sharply increased from day 0 to day 7, especially in the control group, which reached a final value of 0.96 mg MDA/kg (*p* < 0.05) that was significantly higher than the other groups. Its higher TBARS value could be attributed to its higher fat content. However, Trik et al. [35] mentioned that, compared to back-fat, the healthier vegetable oil in the konjac matrix substitute could increase the TBARS value of the dry sausages because of the high quantities of unsaturated (particularly polyunsaturated) fatty acids in the oil. Contrary to this outcome, we prepared CFS using commercial corn germ oil containing a natural antioxidant tocopherol, which protects against oxidation. Valencia et al. [36] claimed that there was no obvious lipid oxidation in dry fermented sausages enriched with fish oil and synthetic antioxidants (butylated hydroxytoluene or butylated hydroxyanisole). The same result was found in Dutch-style low-fat fermented sausages prepared using commercial fish oil instead of pork back-fat [37].

### 3.3. Colour Analysis

Colour is a useful indicator of the quality of meat products, particularly dry fermented sausages, and it often affects consumers’ purchase desire [38]. The colour coordinates *L**, *a** and *b** are presented in Table 4. There was a gradual decrease in lightness (*L**) but an increase in redness (*a**) and yellowness (*b**) for all the samples during the fermentation stage (*p* < 0.05). At the end of fermentation (7 days), *L** was similar between the control group and FS1 treatment (*p* > 0.05) and lowest in the FS4 treatment (*p* < 0.05). This result may be attributed to the continuous water loss during fermentation, causing the colour of the sausages to darken [10]. It was also notable that the FS3 and FS4 treatments, with higher fat replacement levels, had the highest *b** among the samples during the fermentation stage (*p* < 0.05). On the one hand, lipid oxidation may have contributed to the increase in *b**. Liu et al. [39] found a close association between *b** and the yellow pigment produced by the reaction of lipid oxidation products with the amine in protein or phospholipid head groups. On the other hand, it is noteworthy that CFS is more yellow than true back-fat, which may also explain the comparatively higher *b** in the FS3 and FS4 treatments. Additionally, the FS3 and FS4 treatments had a higher *a** relative to the control group at day 7 (*p* < 0.05). Similar results were found in Spanish dry fermented sausages and dry ripened sausages [15,40].

### 3.4. TPA

The TPA of foods is a key indicator to simulate and understand their perceived textural characteristics. The instrumental texture properties obtained using TPA were hardness, chewiness, springiness and resilience, as shown in Table 5. The hardness and chewiness increased sharply in all the samples throughout the fermentation process (*p* < 0.05). These properties were comparable between the FS1 and FS2 treatments (*p* > 0.05) and highest in the FS4 treatment at day 7 (*p* < 0.05), perhaps due to the lower back-fat content and water content in the FS4 treatment. Water content mainly impacts the final texture of fermented sausages and is negatively related to the hardness [41]. CFS is a gelatinous material that has a high water content. As fermentation progressed, the water in the CFS gradually evaporated, increasing the hardness and chewiness of the FS treatments. Similar conclusions have been reported in fat-reduced non-acid fermented sausages and low-fat dry fermented sausages formulated with a konjac gel matrix [2,42]. The springiness and resilience of all the treatments had decreased significantly at the end of fermentation (*p* < 0.05). FS4 displayed the lowest springiness and resilience at day 7 (*p* < 0.05), producing an unsatisfactory texture quality.

### 3.5. E-Nose Analysis

E-nose is a powerful tool that has recently been widely used to distinguish odour profiles between different meat products, such as dry fermented sausages, smoked sausage and golden pompano [25,43,44]. A total of 10 gas sensors were set in the PEN3 system. These sensors were sensitive to aromatic benzenes (W1 C), nitrogen oxides (W5 S), ammonia (W3 C), hydrides (W6 S), short-chain alkane aromatic components (W5 C), methyl compounds (W1 S), sulphides (W1 W), organic sulphides (W2 W), long-chain alkanes (W3 S) and alcohols, aldehydes and ketones (W2 S). As shown in Figure 3A, the W6 S, W1 S and W2 S sensors exhibited the strongest responses to volatile compounds in all the treatments, indicating that hydride constituents, methyl compounds, alcohols, aldehydes and ketones were formed during fermentation. The response values of the W6 S, W1 S and W2 S sensors decreased with the increasing levels of CFS (*p* < 0.05), suggesting that the higher fat replacement levels in the sausages might impede the formation of characteristic flavours.

The PCA loading plot of all the treatments and sensors are shown in Figure 3B. PC1 and PC2 explained 96.0% (83.9% and 12.1%, respectively) of the total variance. Therefore, it is considered that Figure 3B represents the majority of the information [45]. The PC1 spatial regions of each treatment indicated a cluster comprising FS1, FS2 and FS3, suggesting they shared similar flavours [44]. The control group was located to the right and the FS4 treatment was located to the left, both distant from the other samples. The results indicated that CFS could change the flavour of the dry fermented sausages, especially at an 80% fat replacement. The spatial distribution of the E-nose sensors is also shown in Figure 3B. The sensors W1 S, W2 S, W5 S, W6 S and W1 W contributed most to the control group, indicating that the corresponding flavour substances were dominant. In contrast, the sensors W3 C, W5 C and W2 W contributed most to the FS4 treatment, showing that ammonia, organic sulphides and short-chain alkane aromatic components were dominant.

### 3.6. Sensory Analysis

The sensory scores for all the envaulted attributes (colour, texture, juiciness, firmness, flavour and acceptability) and the appearance of the uncooked sausages, cooked sausages and slices of cooked sausages containing true back-fat and CFS are shown in Figure 4 and Figure 5. Atypical appearances were observed in the sausages with high levels of CFS. First of all, the appearance of the sausages with high levels of CFS were more wrinkled and irregular than the others due to the lower moisture content at the end of fermentation. Moreover, the captured image of the uncooked sausages without or with lower levels of CFS presented more white true back-fat than the sausages with high levels of CFS, which highlighted the white appearance of true back-fat compared to CFS. The true back-fat was swollen and shiny after cooking, and the slices of cooked sausage containing true back-fat were relatively compact. It is noted that these atypical appearances (wrinkled appearance and white fat losing) presented in the sausages with high levels of CFS (FS3 and FS4) may lead to a significant reduction in the desire and appetite of consumers. The FS3 and FS4 treatments received the lowest scores for all the attributes (*p* < 0.05), demonstrating that excessive fat replacement may cause a noticeable deterioration of the sensory quality of the sausages. Moreover, consistent with the moisture content and *a*_w_ results, the excessive moisture evaporation led to the dryness and firmness of the FS3 and FS4 treatments [32].

The flavour of Harbin dry sausages is mainly affected by the type and quantity of ingredients. Flavour was comparable among the control, FS1 and FS2 treatments (*p* > 0.05). Therefore, if the level of fat replacement with CFS is controlled within a reasonable range, the flavour will not be remarkably affected. Similar results were reported by Liaros et al. [9].

The juiciness showed a similar behaviour to the flavour, and the lowest scores were contained for the FS4 treatment (*p* < 0.05). No differences were perceived among the control group, FS1 and FS2 treatments (*p* > 0.05). The higher juiciness scores mainly related to the cooked true back-fat. The lowest levels of true back-fat in FS3 and FS4 treatments caused a decrease in the perceived juiciness.

In general, the panellists deemed the FS3 and FS4 treatments limited by harder and drier meat systems compared with the control group. The FS1 and FS2 treatments had acceptable sensory characteristics, meaning that the upper limit of using CFS to replace pork back-fat in Harbin dry fermented sausages is 40%.

### 3.7. HCA Analysis

HCA is the most-used data visualisation tool to assess the multivariate association among treatment groups. The data are depicted in a heatmap in which the higher scores are in red and the lowers score in blue [46]. As shown in Figure 6, cluster one indicated that the control, FS1 and FS2 treatments were associated with high resilience, moisture content, *L**, TBARS value, springiness and *a*_w_, opposite to the FS3 and FS4 treatments. In cluster two, the evident downward trend of *b**, *a**, pH, chewiness and hardness in the control, FS1 and FS2 treatments contrasted with the upward trend of these properties in the FS3 and FS4 treatments. These results showed that indicators in the same cluster had significant positive correlations with each other and were negatively correlated with those in different clusters. For instance, the higher moisture content in the sausages normally meant higher springiness and lower hardness. Notably, the FS1 and FS2 treatments were grouped with the control group, and the FS3 treatment was grouped with the FS4 treatment, which indicated that an excessive fat replacement with CFS might induce noticeable changes in the quality characteristics of traditional full-fat dry fermented sausages.

## 4. Conclusions

In summary, from the results of the various analytical techniques, the partial replacement of back-fat with CFS was a feasible way to reduce the fat content of Harbin dry fermented sausages, depending on the replacement level. Compared with the control group, replacing 20 and 40% of the back-fat with CFS did not affect the physical and sensory characteristics of the sausage. However, when the replacement level reached 60% or more, the physical and sensory characteristics changed, as evidenced by the decreased moisture content and *a*_w_ and the increased pH, hardness, chewiness and atypical appearance of the sausages. Additionally, E-nose analysis and HCA analysis affirmed the negative correlations between the control and the FS4 (80% back-fat replacement) treatment in quality characteristics. It is noted that this study provides another option for food producers to replace animal with plant-based ingredients. Replacing up to 40% of animal fat with plant-based CFS can reduce the environmental footprint of Harbin dry sausages, make them healthier for consumers and more ethical without significantly affecting their sensory or physical characteristics. Further research will focus on the following two aspects: (1) how to effectively retard the moisture evaporation of the CFS, which could probably improve the fat replacement levels, as well as promote quality characteristics of Harbin dry sausages; (2) optimising the technological conditions to improve the production technology and quality of low-fat Harbin dry fermented sausages.

## Figures and Tables

**Figure 1 foods-10-01460-f001:**
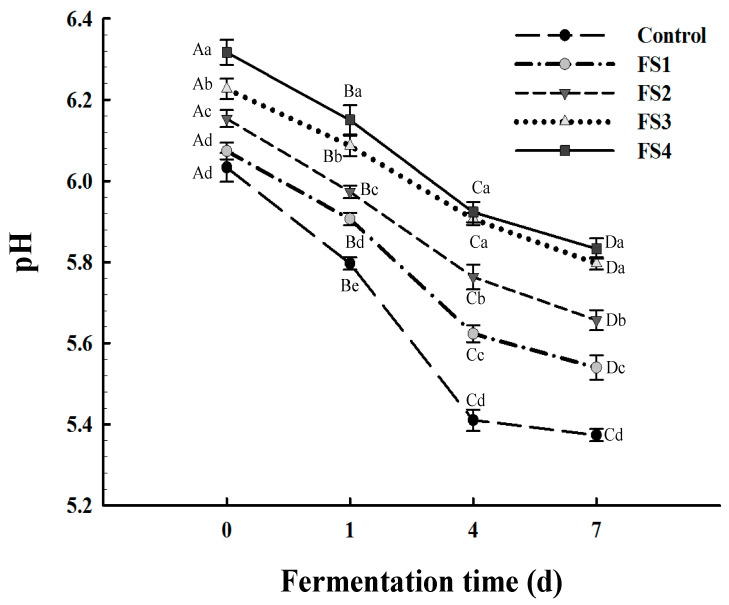
Changes in pH of Harbin dry sausages with different replacement levels of CFS during fermentation. Control: 100% backfat; FS1: 80% backfat and 20% CFS; FS2: 60% backfat and 40% CFS; FS3: 40% backfat and 60% CFS; FS4: 20% backfat and 80% CFS. Different lowercase letters (a–d) mean significant differences among the treatments (*p* < 0.05). Different uppercase letters (A–D) mean significant differences among different fermentation times (*p* < 0.05).

**Figure 2 foods-10-01460-f002:**
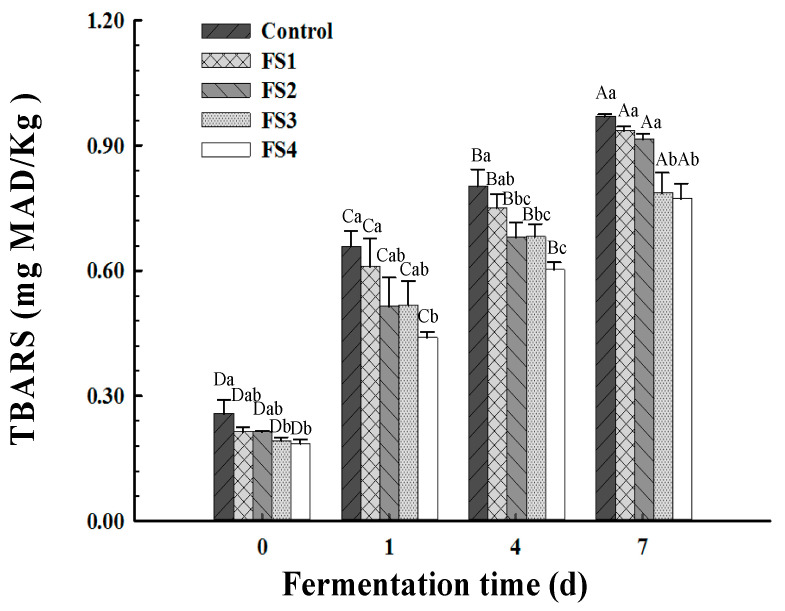
Changes in thiobarbituric acid reactive substance (TBARS) values of Harbin dry sausages with different replacement levels of CFS during fermentation. Control: 100% backfat; FS1: 80% backfat and 20% CFS; FS2: 60% backfat and 40% CFS; FS3: 40% backfat and 60% CFS; FS4: 20% backfat and 80% CFS. Different lowercase letters (a–c) mean significant differences among the treatments (*p* < 0.05). Different uppercase letters (A–D) mean significant differences among different fermentation times (*p* < 0.05).

**Figure 3 foods-10-01460-f003:**
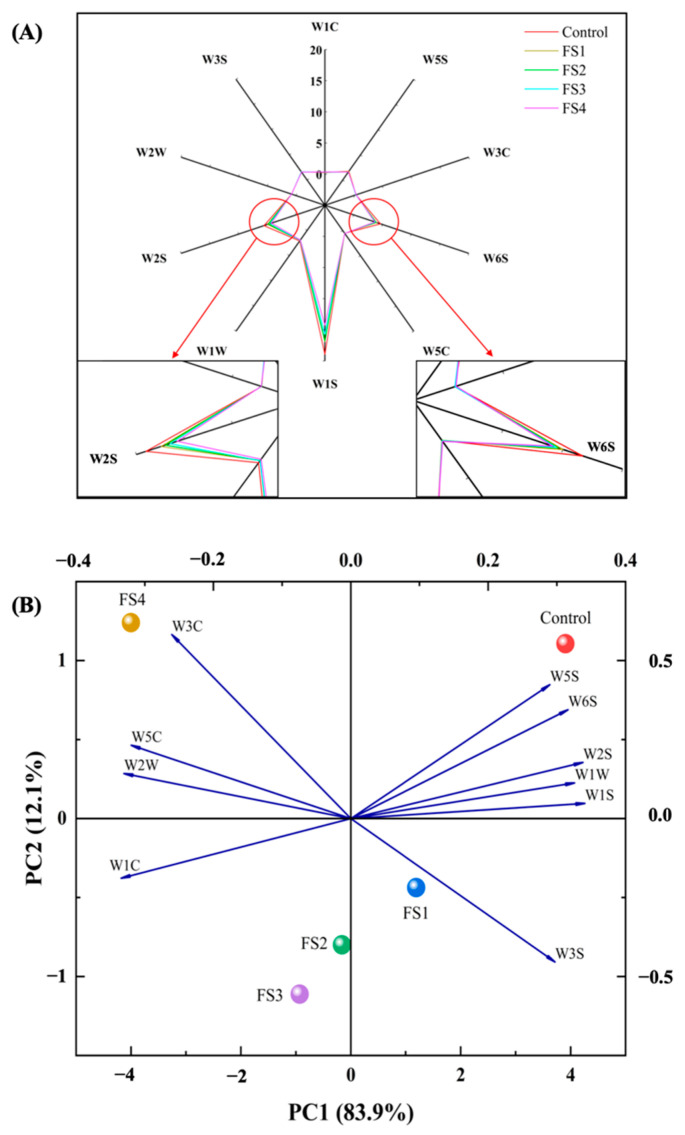
Radar chart of the electronic nose (E-nose) response data (**A**), principal component analysis loading plot of different treatments and sensors (**B**). Control: 100% backfat; FS1: 80% backfat and 20% CFS; FS2: 60% backfat and 40% CFS; FS3: 40% backfat and 60% CFS; FS4: 20% backfat and 80% CFS.

**Figure 4 foods-10-01460-f004:**
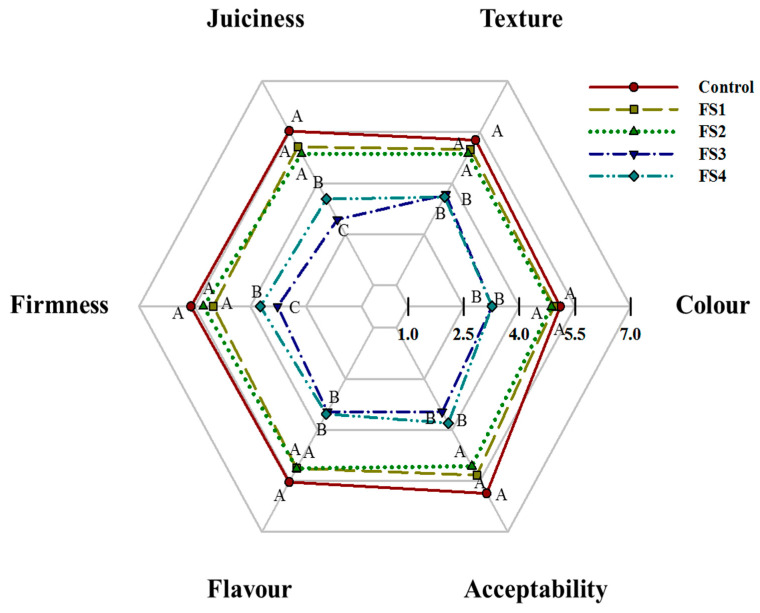
Sensory evaluation of the Harbin dry sausages with different replacement levels of CFS after fermentation. Control: 100% backfat; FS1: 80% backfat and 20% CFS; FS2: 60% backfat and 40% CFS; FS3: 40% backfat and 60% CFS; FS4: 20% backfat and 80% CFS. Different uppercase letters (A–C) mean significant differences among the treatments (*p* < 0.05).

**Figure 5 foods-10-01460-f005:**
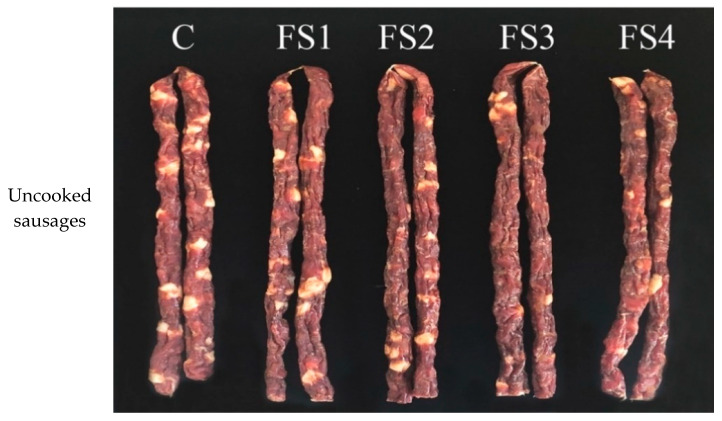
Appearance of the uncooked sausages, cooked sausages and slices of cooked sausages containing true fat and CFS.

**Figure 6 foods-10-01460-f006:**
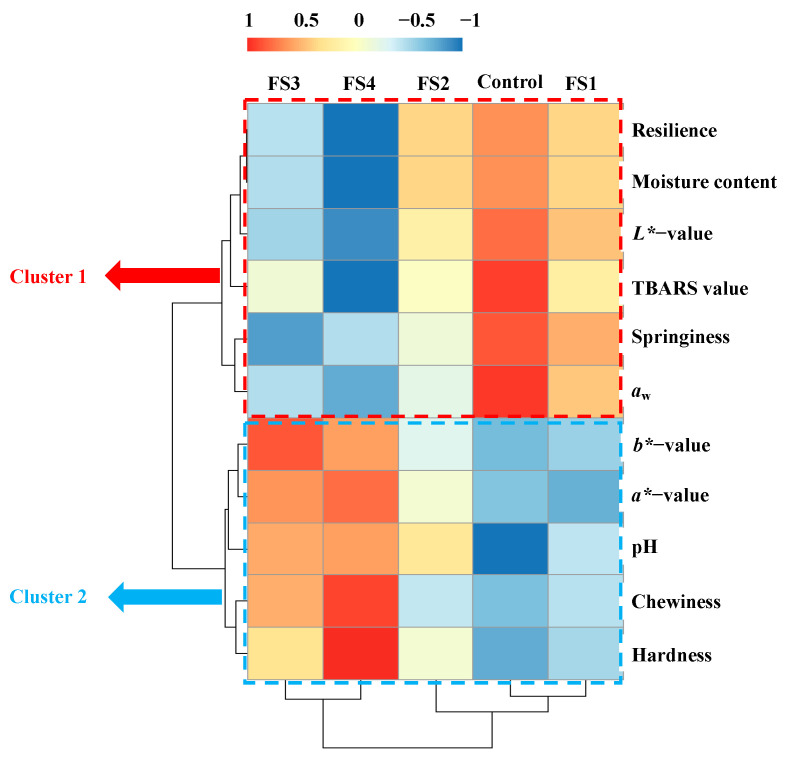
Hierarchical cluster analysis (HCA) of Harbin dry sausages with different replacement levels of CFS after fermentation. Control: 100% backfat; FS1: 80% backfat and 20% CFS; FS2: 60% backfat and 40% CFS; FS3: 40% backfat and 60% CFS; FS4: 20% backfat and 80% CFS.

**Table 1 foods-10-01460-t001:** Formulation (g/kg) of Harbin dry fermented sausages.

Treatments	Lean Meat	Back Fat	CFS
Control	900	100	-
FS1	900	80	20
FS2	900	60	40
FS3	900	40	60
FS4	900	20	80

Control, the dry fermented sausages without CFS; FS1–FS4, the dry fermented sausages containing different replacement levels of CFS (20, 40, 60 and 80 g/kg).

**Table 2 foods-10-01460-t002:** Information of 10 sensors for electronic nose.

Sensors	Representative Material Species	Description
W1 C	Aromatic compounds	Sensitive to aromatic constituents, benzenes
W5 S	Broad range	Sensitive to nitrogen oxides
W3 C	Aromatic	Sensitive to aroma, ammonia
W6 S	Hydrogen	Sensitive to hydrides
W5 C	Arom-aliph	Sensitive to short-chain alkane aromatic components
W1 S	Broad-methane	Sensitive to methyl compounds
W1 W	Sulphur-organic	Sensitive to sulphides
W2 S	Broad-alcohol	Sensitive to alcohols, aldehydes and ketones
W2 W	Sulph-chlor	Sensitive to organic sulphides
W3 S	Methane-aliph	Sensitive to long-chain alkanes

**Table 3 foods-10-01460-t003:** Changes in moisture content (%) and water activity (*a*_w_) of Harbin dry sausages with different replacement levels of CFS during fermentation.

		Fermentation Time (d)			
		0	1	4	7
moisture content (%)	Control	62.52 ± 0.05 ^Ad^	53.39 ± 0.32 ^Ba^	38.37 ± 0.38 ^Ca^	28.57 ± 0.45 ^Da^
	FS1	63.32 ± 0.38 ^Ac^	52.06 ± 0.28 ^Bb^	37.12 ± 0.31 ^Ca^	27.31 ± 0.34 ^Da^
	FS2	64.35 ± 0.25 ^Ab^	50.50 ± 0.38 ^Bc^	35.63 ± 0.43 ^Cb^	27.55 ± 0.39 ^Da^
	FS3	64.88 ± 0.07 ^Ab^	48.35 ± 0.31 ^Bd^	32.64 ± 0.35 ^Cc^	24.36 ± 0.64 ^Db^
	FS4	65.53 ± 0.18 ^Aa^	47.83 ± 0.22 ^Bd^	30.32 ± 0.74 ^Cd^	22.06 ± 0.46 ^Dc^
water activity (*a*_w_)	Control	0.95 ± 0.01 ^Ab^	0.92 ± 0.01 ^Ba^	0.87 ± 0.01 ^Ca^	0.84 ± 0.01 ^Da^
	FS1	0.96 ± 0.01 ^Aab^	0.90 ± 0.01 ^Bb^	0.86 ± 0.01 ^Ca^	0.82 ± 0.01 ^Db^
	FS2	0.96 ± 0.01 ^Aa^	0.91 ± 0.01 ^Bb^	0.84 ± 0.01 ^Cb^	0.81 ± 0.01 ^Db^
	FS3	0.96 ± 0.01 ^Aa^	0.90 ± 0.01 ^Bbc^	0.83 ± 0.01 ^Cc^	0.79 ± 0.01 ^Dc^
	FS4	0.97 ± 0.01 ^Aa^	0.90 ± 0.01 ^Bc^	0.80 ± 0.01 ^Cd^	0.78 ± 0.01 ^Dd^

^a–d^ Means within the same column for the same index with different lowercase letters differ significantly among the treatments (*p* < 0.05). ^A–D^ Means within the same row with different uppercase letters differ significantly among different fermentation times (*p* < 0.05). Control: 100% backfat; FS1: 80% backfat and 20% CFS; FS2: 60% backfat and 40% CFS; FS3: 40% backfat and 60% CFS; FS4: 20% backfat and 80% CFS.

**Table 4 foods-10-01460-t004:** Changes in colour values (*L**, *a** and *b**) of Harbin dry sausages with different replacement levels of CFS during fermentation.

		Fermentation Time (d)			
		0	1	4	7
*L**-value	Control	61.30 ± 1.01 ^Aa^	50.42 ± 0.71 ^Ba^	37.51 ± 1.29 ^Ca^	31.74 ± 0.61 ^Da^
	FS1	60.91 ± 0.70 ^Aa^	47.03 ± 0.25 ^Bb^	33.58 ± 0.71 ^Cb^	31.71 ± 0.36 ^Da^
	FS2	60.58 ± 0.54 ^Aa^	45.68 ± 0.21 ^Bc^	32.54 ± 0.62 ^Cbc^	29.92 ± 0.58 ^Db^
	FS3	60.69 ± 0.60 ^Aa^	43.68 ± 0.16 ^Bd^	30.40 ± 0.82 ^Ccd^	28.59 ± 0.66 ^Db^
	FS4	60.28 ± 0.39 ^Aa^	43.27 ± 0.18 ^Bd^	29.49 ± 0.46 ^Cd^	26.81 ± 0.51 ^Dc^
*a**-value	Control	10.11 ± 0.04 ^Cd^	12.84 ± 0.04 ^Bb^	14.13 ± 0.14 ^Ac^	14.42 ± 0.12 ^Ac^
	FS1	10.23 ± 0.05 ^Cd^	13.04 ± 0.10 ^Bb^	14.29 ± 0.13 ^Ac^	14.43 ± 0.22 ^Ac^
	FS2	10.52 ± 0.14 ^Cc^	13.41 ± 0.49 ^Bab^	14.77 ± 0.11 ^ABb^	14.96 ± 0.07 ^Ab^
	FS3	12.07 ± 0.06 ^Cb^	14.00 ± 0.15 ^Ba^	15.62 ± 0.20 ^Aa^	15.77 ± 0.12 ^Aa^
	FS4	12.34 ± 0.04 ^Ca^	13.97 ± 0.05 ^Ba^	15.84 ± 0.14 ^Aa^	15.95 ± 0.10 ^Aa^
*b**-value	Control	14.04 ± 0.10 ^Cc^	14.07 ± 0.28 ^Cc^	14.28 ± 0.12 ^Bd^	14.88 ± 0.03 ^Ac^
	FS1	14.17 ± 0.15 ^Cbc^	14.15 ± 0.17 ^Cbc^	14.41 ± 0.08 ^Bc^	14.95 ± 0.07 ^Ac^
	FS2	14.26 ± 0.23 ^Cb^	14.31 ± 0.20 ^Cb^	14.66 ± 0.18 ^Bb^	15.13 ± 0.12 ^Ab^
	FS3	14.67 ± 0.08 ^Ca^	14.97 ± 0.03 ^Ba^	14.92 ± 0.10 ^Ba^	15.83 ± 0.11 ^Aa^
	FS4	14.64 ± 0.22 ^Ca^	14.96 ± 0.13 ^Ba^	14.86 ± 0.04 ^Ba^	15.67 ± 0.23 ^Aa^

^a–d^ Means within the same column for the same index with different lowercase letters differ significantly among the treatments (*p* < 0.05). ^A–D^ Means within the same row with different uppercase letters differ significantly among different fermentation times (*p* < 0.05). Control: 100% backfat; FS1: 80% backfat and 20% CFS; FS2: 60% backfat and 40% CFS; FS3: 40% backfat and 60% CFS; FS4: 20% backfat and 80% CFS.

**Table 5 foods-10-01460-t005:** Changes in texture profile analysis (TPA) of Harbin dry sausages with different replacement levels of CFS during fermentation.

Fermentation Time (d)	Treatments	Hardness (N)	Springiness	Chewiness (N)	Resilience
0	Control	13.61 ± 1.07 ^Ba^	0.63 ± 0.02 ^Aa^	4.45 ± 0.37 ^Ba^	0.23 ± 0.01 ^Aa^
	FS1	13.98 ± 0.71 ^Ba^	0.62 ± 0.05 ^Aa^	4.34 ± 0.31 ^Ba^	0.20 ± 0.01 ^Aab^
	FS2	12.71 ± 0.68 ^Bab^	0.61 ± 0.06 ^Aab^	4.15 ± 0.22 ^Bab^	0.19 ± 0.01 ^Ab^
	FS3	12.28 ± 0.63 ^Bab^	0.58 ± 0.04 ^Ab^	3.73 ± 0.27 ^Bb^	0.17 ± 0.01 ^Ac^
	FS4	11.16 ± 0.84 ^Bb^	0.58 ± 0.05 ^Ab^	3.47 ± 0.17 ^Bb^	0.16 ± 0.01 ^Ad^
7	Control	60.73 ± 1.17 ^Ac^	0.59 ± 0.04 ^Ba^	17.99 ± 0.13 ^Ac^	0.17 ± 0.01 ^Ba^
	FS1	61.66 ± 0.67 ^Ac^	0.58 ± 0.03 ^Ba^	18.53 ± 0.39 ^Ab^	0.16 ± 0.01 ^Ba^
	FS2	63.17 ± 1.16 ^Abc^	0.56 ± 0.04 ^Bb^	18.65 ± 0.53 ^Ab^	0.16 ± 0.01 ^Ba^
	FS3	64.66 ± 1.34 ^Aab^	0.54 ± 0.07 ^Bc^	21.24 ± 0.67 ^Aa^	0.13 ± 0.01 ^Bb^
	FS4	67.34 ± 0.95 ^Aa^	0.55 ± 0.05 ^Bc^	22.32 ± 0.41 ^Aa^	0.11 ± 0.01 ^Bc^

^a–d^ Means within the same column for the same fermentation time with different lowercase letters differ significantly among the treatments (*p* < 0.05). ^A–B^ Means within the same column for the same treatment with different uppercase letters differ significantly among different fermentation times (*p* < 0.05). Control: 100% backfat; FS1: 80% backfat and 20% CFS; FS2: 60% backfat and 40% CFS; FS3: 40% backfat and 60% CFS; FS4: 20% backfat and 80% CFS.

## Data Availability

The data presented in this study are available in the article.

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
