# Peer review of "Composite Gel Fabricated with Konjac Glucomannan and Carrageenan Could Be Used as a Cube Fat Substitute to Partially Replace Pork Fat in Harbin Dry Sausages"

_foods, 2021, doi:10.3390/foods10071460_

Round 1

Reviewer 1 Report

Overall, a very detailed paper with excellent figures representing the significant results. The authors could do more to highlight the purpose of replacing animal fat with plant-based CFS, thus demonstrating the value of this work. I have some suggestions and questions below, but overall this is a useful and well-written manuscript.

Introduction

Line 50: You can add more about the benefits of reducing animal products in favour of plant-based products. Your proposed products with CFS replacing animal fat will presumably (a) have a lower environmental impact, since animal products are very resource-intensive, (b) lowers the public health risks of animal farming, e.g. pandemics and antibiotic resistance, and (c) represent progress in animal ethics, given fewer animals reared for the same amount of sausage. You could expand more on these benefits to highlight the many other rationales for replacing pork fat with CFS.

Materials and Methods

Line 95: The authors might note, ‘Research involving human or animal participants usually requires ethical approval, but this protection is not extended to farmed animals.’

Line 152: How were the 30 tasters recruited? Were they paid? Was ethical or IRB approval required or sought for the 30 tasters? (This may be provided by the panel)

Results and discussion

Line 197: If the CFS was degraded due to the specific fermentation process associated with Harbin sausage, could it be useful in other meat products which do not go through fermentation?

Line 254: It is notable that the measurements of FS3 and FS4 at day 4 are similar to the control at day 1; FS3/4 at day 7 are similar to the control at day 4, etc. - does this suggest that producers could reduce the animal fat content and compensate by fermenting the product for longer, thus still achieving high levels of these characteristics?

Line 369: I am a fan of radar plots to represent this kind of data. Very clear.

Conclusions

You should add a final few sentences about the broader impact/purpose of this work, something along the lines of ‘Replacing up to 40% of animal fat with plant-based CFS can reduce the environmental footprint of Harbin sausage, make it healthier to consumers, and more ethical without significantly affecting its sensory or physical characteristics. Food producers should investigate further options for replacing animal ingredients with plant-based to achieve similar efficiency gains.’

Author Response

Response to Reviewer 1#

 Q1: Line 50: You can add more about the benefits of reducing animal products in favour of plant-based products. Your proposed products with CFS replacing animal fat will presumably (a) have a lower environmental impact, since animal products are very resource-intensive, (b) lowers the public health risks of animal farming, e.g. pandemics and antibiotic resistance, and (c) represent progress in animal ethics, given fewer animals reared for the same amount of sausage. You could expand more on these benefits to highlight the many other rationales for replacing pork fat with CFS.

A1: This is a good suggestion. The reviewer is right. These plant-based ingredients can be used as fat substitutes may be due to three main benefits. First of all, plant-based ingredients are rich in dietary fiber and has no caloric value, more conducive to the health of consumers. Secondly, using the plant-based ingredients instead of animal fat is beneficial for resource-intensive meat products manufacturers to reduce the environmental problems. Thirdly, and perhaps most importantly, the plant-based ingredients own high water retention capacity which can facilitate the regular moisture release occurring during the fermentation process, ensuring sensory property of fermented dry sausages. According to the reviewer’s opinion, we have added some statements as follows:

“Several studies have attempted to overcome these issues by using plant-based ingredients formed from vegetable oils and typical neutral polysaccharide gels (such as konjac gel, carrageenan, carboxymethylcellulose, jerusalem artichoke powder, locust bean gum and xanthan gum) to partially reduce the fat content in these meat products [2,11,12]. The plant-based ingredients are rich in dietary fiber and has no caloric value, more conducive to the health of consumers [13]. Using the plant-based ingredients instead of animal fat may be beneficial for resource-intensive meat product manufacturers to reduce the environmental problems. More importantly, the plant-based ingredients own high water retention capacity which can facilitate the regular moisture release occurring during the fermentation process, ensuring sensory property of fermented dry sausages [14].”

The two newly added references:

[13] Al-Ghazzewi, F.H.; Khanna, S.; Tester, R.F.; Piggott, J. The potential use of hydrolysed konjac glucomannan as a prebiotic. J. Sci. Food Agr. 2007, 87, 1758-1766. https://doi.org/10.1002/jsfa.2919

[14] Wang, X.X.; Xie, Y.Y.; Li, X.M.; Liu, Y.; Yan, W.J. Effects of partial replacement of pork back fat by a camellia oil gel on certain quality characteristics of a cooked style Harbin sausage. Meat Sci. 2018, 146, 154-159. https://doi.org/10.1016/j.meatsci.2018.08.011

We are indebted to the reviewer for this constructive suggestion to improve the quality of the manuscript. Thanks.

Q2: Line 95: The authors might note, ‘Research involving human or animal participants usually requires ethical approval, but this protection is not extended to farmed animals.’ 

A2: This is a good suggestion. According to the reviewer’s opinion, we have added the statement as follows:

“Fresh pork lean-meat and back-fat were obtained from the Carrefour supermarket (Harbin, China) and transported on ice to the meat science laboratory, although some researches which involving human or animal participants usually requires ethical approval, this protection is not extended to farmed animals. The visible fat and connective tissue on the lean pork were removed. Ingredients used to prepare CFS included corn germ oil (COFCO Co., Ltd., Heilongjiang, China), konjac flour (glucomannan 83%, 120 mesh; Johnson Konjac Technology Co., Ltd., Hubei, China), κ-carrageenan flour (Jingxie Marine Technology Co., Ltd., Shandong, China), barley β-glucan (Yuwei Bio-technology Co., Ltd., Beijing, China) and food-grade Na2CO3 (Zhenlemen Food Co., Ltd., Jiangsu, China). Other additives, such as NaCl, sodium nitrite and flavouring, were also purchased from the Carrefour supermarket.”

We are indebted to the reviewer for this constructive suggestion to improve the quality of the manuscript. Thanks.

Q3: Line 152: How were the 30 tasters recruited? Were they paid? Was ethical or Institutional review board IRB approval required or sought for the 30 tasters? (This may be provided by the panel)

A3: This is a good question. First of all, the 30 tasters were selected in the meat science laboratory of Northeast Agricultural University (Harbin, China). They are professional researchers in the field of meat products processing. All of the tasters are volunteers, and don't need to be paid. Secondly, this study was registered and approved by the Ethics in Research Committee of Northeast Agricultural University (Harbin, China), and the procedures of the sensory analysis were implemented according to the ISO standards. In our revised manuscript, we have added some statements as follows:

“This study was registered and approved by the Ethics in Research Committee of Northeast Agricultural University (Harbin, China). The procedure for sensory evaluation was adopted from Kong et al. [26] with some modifications. Thirty sensory analysis panellists (15 females and 15 males) were selected in the meat science laboratory of Northeast Agricultural University due to their experience in sensory evaluation of meat products. All of the panellists signed a consent form agreeing to participate as volunteers in the sensory analysis. Before the sensory evaluation, the experts from the meat science laboratory of Northeast Agricultural University conducted three preliminary, sample familiarisation training sessions for the 30 panellists, and a "warm-up" sample (cooked dry sausage slices) was submitted to every panellist to evaluate each sensory trait. Afterwards, the steamed sausage samples of each treatment were sliced into approximately 2 mm-thick pieces to begin the evaluation. The procedures of the sensory analysis were implemented according to a seven-point scale method [27], and carried out in a sensory laboratory designed in accordance with International Standard Organisation (ISO) [28].”

The two newly added references:

[27] International Standard Organisation (ISO). Available online: https://www.iso.org/standard/50125.html (accessed on 01 December 2014).

[28] International Standard Organisation (ISO). Available online: https://www.iso.org/standard/36385.html (accessed on 01 December 2007).

We are indebted to the reviewer for this constructive suggestion to improve the quality of the manuscript. Thanks.

Q4: Line 197: If the CFS was degraded due to the specific fermentation process associated with Harbin sausage, could it be useful in other meat products which do not go through fermentation?

A4: This is a good question. The reviewer is right. The production principle of fermented sausages is different from that of non-fermented sausages. The high levels of CFS may increase the product hardness due to higher water loss during fermentation. However, the water loss could not induce degradation of CFS in the fermentation process of dry sausage. Moreover, in another published paper (Science and Technology of Food Industry, 2021, 42(9): 85-93, in Chinese), the CFS were used in red sausage (a type of non-fermented sausage) production to replace pork fat with 20%, 40%, 60% and 80%, respectively. The results showed that CFS could partially replace pork fat in red sausage, and the optimal replacement ratio was 60%. In our revised manuscript, we have added some statements as follows:

“The same results were also reported in low-fat dry fermented sausages amended with gelatinous substances [2,17]. Moreover, the water loss could not induce degradation of CFS in the fermentation process of dry fermented sausages, which made it could be potentially used in non-fermented sausages.”

We are indebted to the reviewer for this constructive suggestion to improve the quality of the manuscript. Thanks.

Q5: Line 254: It is notable that the measurements of FS3 and FS4 at day 4 are similar to the control at day 1; FS3/4 at day 7 are similar to the control at day 4, etc. - does this suggest that producers could reduce the animal fat content and compensate by fermenting the product for longer, thus still achieving high levels of these characteristics?

A5: This is a good question. The reviewer is right, the measurements of TBARS values of FS3 and FS4 at day 4 are similar to that of the control at day 1, and the TBARS values of FS3/4 at day 7 are similar to that of the control at day 4. However, the physicochemical and sensory characteristics of the Harbin dry sausages not only depend on the TBARS values, also by other the influence of various factors. For instance, moisture content and aw gradually decreased in all treatments during the 7-day fermentation process, and the FS3/4 displayed significantly lower moisture contents and aw compared to the control at the end of the fermentation process. The high levels of CFS may increase the product hardness due to higher water loss during fermentation. If reduce the animal fat content and compensate by fermenting the product for longer, the sensory quality of fermented sausage will not be recognized by consumers. Thus, due to some possible deterioration of the quality, extend the fermentation time can not replace more animal fat in the production of dry fermented sausages.

Q6: Line 369: I am a fan of radar plots to represent this kind of data. Very clear.

A6: Thanks for your praise. We will do our best in our future work.

Q7: You should add a final few sentences about the broader impact/purpose of this work, something along the lines of ‘Replacing up to 40% of animal fat with plant-based CFS can reduce the environmental footprint of Harbin sausage, make it healthier to consumers, and more ethical without significantly affecting its sensory or physical characteristics. Food producers should investigate further options for replacing animal ingredients with plant-based to achieve similar efficiency gains.’

A7: This is a good suggestion. According to the reviewer’s opinion, we have added some sentences to explain the broader impact/purpose of this work as follows:

“Additionally, E-nose analysis and HCA analysis affirmed the negative correlations between the control and FS4 (80% back-fat replacement) treatment in quality characteristics. It is noted that this study provides another option for food producers to replace animal ingredients with plant-based. Replacing up to 40% of animal fat with plant-based CFS can reduce the environmental footprint of Harbin dry sausage, make it healthier to consumers, and more ethical without significantly affecting its sensory or physical characteristics.”

We are indebted to the reviewer for this constructive suggestion to improve the quality of the manuscript. Thanks.

Reviewer 2 Report

This article analysis the possibility of reducing fat in Harbin dry sausages by substitution of back fat by Konjac Glucomannan and Carrageenan gels at different replacement levels. The article is very well written and is of interest both to industry and consumers due to the trend and need of reduction in the consumption of saturated fat. Just some few minor points needs to be addressed:

Authors state both in the abstract and conclusion about an atypical appearance, however this is not present in the discussion and also from the image provided their seems to be no atypical appearance, so could author please explain in more details what they means by this?

Authors should also give a quick explanation as of to why Konjac Glucomannan and Carrageenan was chosen over  the other mentioned polysaccharides (carboxymethylcellulose, jerusalem artichoke powder, locust 65 bean gum and xanthan gum).

In statistical analysis in the methodology authors should mention they carried out a PCA analysis

Line 124: Include a bit more details about the TBARS analysis.

Line 140: Define TPA in the title

Line 173: Define HCA at first use

Line 382: Define HCA in the tittle

Author Response

Response to Reviewer 2#

 Q1: Authors state both in the abstract and conclusion about an atypical appearance, however this is not present in the discussion and also from the image provided their seems to be no atypical appearance, so could author please explain in more details what they means by this?

 A1: This is a good suggestion. First of all, we were apologized for the inappropriate statement in our initial manuscript. The atypical appearance is mainly reflected in two points. On one hand, the appearance of sausages with high levels of CFS were more wrinkled and irregular than the others due to the lower moisture content at the end of fermentation. On the other hand, the sausages with high levels of CFS presented few white true back-fat. More importantly, the true back-fat was swollen and shiny after cooking, and the slices of cooked sausage containing true back-fat were relatively compact. These atypical appearances may lead to a significant reduction in desire and appetite of consumers. According to the reviewer’s opinion, we have added some statements to define the atypical appearance under the section of “3.6. Sensory analysis” as follows:

“Atypical appearances were observed in the sausages with high levels of CFS. First of all, the appearance of sausages with high levels of CFS were more wrinkled and irregular than the others for the lower moisture content at the end of fermentation. Moreover, the captured image of uncooked sausages without or with lower levels of CFS presented more white true back-fat than the sausages with high levels of CFS, which highlighted the white appearance of true back-fat compared to CFS. The true back-fat was swollen and shiny after cooking, and the slices of cooked sausage containing true back-fat were relatively compact. It is noted that these atypical appearances (wrinkled appearance and white fat losing) presented in the sausages with high levels of CFS (FS3 and FS4) may lead to a significant reduction in desire and appetite of consumers.”

We are indebted to the reviewer for this constructive suggestion to improve the quality of the manuscript. Thanks.

 Q2: Authors should also give a quick explanation as of to why Konjac Glucomannan and Carrageenan was chosen over the other mentioned polysaccharides (carboxymethylcellulose, jerusalem artichoke powder, locust bean gum and xanthan gum).

 A2: This is a good suggestion. Konjac glucomannan is a neutral polysaccharide extracted from the amorphophallus konjac, a native plant of East Asia. Its use as a food additive is authorized in area normalization (GBT 18104-2000) of the Standardization Administration of the People's Republic of China (SAC). Several studies had proved that it forms gels which combined with carrageenan can be successfully used as ‘fat analogs’ in the formulation of low-fat meat products (Meat Sci. 2012, 92: 144-150). In our previous study, the best physicochemical and sensory properties were found in the gel cube fat substitutes (CFS) prepared by konjac glucomannan and κ-carrageenan, and the CFS conferred sensory properties of juiciness and texture similar to that of fat (Food Research & Development, 2018, 39, 12-18, in Chinese). Thus, we optimized the composition of CFS as a practical and effective option to partially reduce the fat content in Harbin dry sausages. According to the reviewer’s opinion, we have added some statements in the section of “Introduction” as follows:

“Konjac glucomannan is a neutral polysaccharide extracted from the amorphophallus konjac, a native plant of East Asia. Several studies had proved that it forms gels which combined with carrageenan can be successfully used as ‘fat analogs’ in the formulation of low-fat meat products [2]. Moreover, in our previous study, the best physicochemical and sensory properties were found in the gel cube fat substitutes (CFS) prepared by konjac glucomannan and κ-carrageenan, and the CFS conferred sensory properties of juiciness and texture similar to that of fat [18]”

We are indebted to the reviewer for this constructive suggestion to improve the quality of the manuscript. Thanks.

 Q3: In statistical analysis in the methodology authors should mention they carried out a PCA analysis

 A3: This is a good suggestion. According to the reviewer’s opinion, we have added a statement about the PCA analysis under the section of “2.10” as follows:

“Principal component analysis (PCA) was performed among the treatments and sensors of E-nose analysis using SPSS Statistics version 22.0 (Analytical Software, New York, USA)”

We are indebted to the reviewer for this constructive suggestion to improve the quality of the manuscript. Thanks.

 Q4: Line 124: Include a bit more details about the TBARS analysis.

 A4: This is a good suggestion. According to the reviewer’s opinion, we have added more details about the TBARS analysis under the section of “2.5” as follows:

“Lipid oxidation was evaluated by measuring the TBARS using the method of Wen et al. [23]. Briefly, 2.0 g of minced sausage samples were weighed and mixed with 3.0 mL of thiobarbituric acid, followed by addition of 17.0 mL 2.5% trichloroacetic acid. Then, the mixture was heated in a boiling water for 30 min and cooled at room temperature. After that, 4.0 mL of suspension was mixed with the same volume of chloroform, and then centrifuged at 1,800 × g for 10 min. The supernatant was determined at 532 nm. Results were expressed as milligrams of malonaldehyde (MDA) per kilogram of sausage, and calculated using the following equation: TBARS (mg/kg) = A532 / ω × 9.48

Among the equation: A532 is the absorbance (532 nm) of the assay solution, ω is the sample weight (g), and ‘9.48’ is a constant derived from the dilution factor and the molar extinction coefficient (152,000 M-1 cm-1) of the red thiobarbituric acid reaction product.”

We are indebted to the reviewer for this constructive suggestion to improve the quality of the manuscript. Thanks.

Q5: Line 140: Define TPA in the title

A5: We have revised it as suggested. Thanks.

Q6: Line 173: Define HCA at first use

A6: We have revised it as suggested. Thanks.

Q7: Line 382: Define HCA in the tittle

A7: We have revised it as suggested. Thanks.